# Impaction Predictors and Diagnostic Performance of CBCT Versus Panoramic Radiography for Supernumerary Teeth in a Romanian Multicenter Cohort

**DOI:** 10.3390/diagnostics15233019

**Published:** 2025-11-27

**Authors:** Cristina Păcurar, Octavia Mesaroș, Andreea Angela Ștețiu, Sorana Maria Bucur, Cristina Nicoleta Mihai, Mariana Păcurar

**Affiliations:** 1Doctoral School, George Emil Palade University of Medicine, Pharmacy, Science, and Technology of Târgu Mureș, 38 Ghe. Marinescu Street, 540142 Târgu Mureș, Romania; cristina.pacurar18@yahoo.com; 2Department of Orthodontics, Faculty of Dentistry, George Emil Palade University of Medicine, Pharmacy, Science, and Technology of Târgu Mureș, 38 Ghe. Marinescu Street, 540142 Târgu Mureș, Romania; uplearning98@gmail.com (O.M.); mariana.pacurar@umfst.ro (M.P.); 3Periodontology Department, Faculty of Medicine, Lucian Blaga University of Sibiu, 550169 Sibiu, Romania; 4Department of Dentistry, Faculty of Medicine, Dimitrie Cantemir University of Târgu Mureș, 540545 Târgu Mureș, Romania; 5Private Dental Practice Bucharest, 012056 Bucharest, Romania

**Keywords:** supernumerary teeth, mesiodens, impaction, cone-beam computed tomography, morphology, malocclusion, eruption disturbance, dento-maxillary anomalies, diagnostic accuracy, epidemiology

## Abstract

**Background:** Supernumerary teeth (ST) are developmental anomalies that may interfere with eruption, alignment, and occlusal balance. Their etiopathogenesis and management remain controversial. This multicentric study aimed to evaluate the epidemiological, morphological, and radiographic features of ST in a Romanian population and identify impact predictors. **Methods:** Between January 2020 and March 2025, 153 consecutive patients (91 males, 62 females; mean age 14.8 ± 6.2 years) with clinically and radiographically confirmed supernumerary teeth were evaluated across three Romanian academic centers: the University Dental Clinic, George Emil Palade University of Medicine, Târgu Mureș (*n* = 78 patients); the Department of Periodontology, Lucian Blaga University of Sibiu (*n* = 45 patients); and the Department of Dentistry, Dimitrie Cantemir University of Târgu Mureș (*n* = 30 patients). **Results:** A total of 185 ST were recorded, most frequently conical (48.6%) and mesiodens (56.2%). Complications were observed in 40.5% of patients. Multivariable analysis identified Angle Class III malocclusion (OR = 1.89; *p* = 0.039) and tuberculate morphology (OR = 2.93; *p* = 0.021) as the strongest independent predictors of impaction, alongside associations with younger age (<13 years) (OR = 3.12; *p* = 0.003) and male gender (OR = 1.78; *p* = 0.046). CBCT demonstrated high diagnostic concordance with OPG (κ = 0.89), but showed superior performance for complex cases, identifying 11 root resorptions and 9 vestibulo-palatal displacements that OPG missed. Multivariable analysis identified Angle Class III malocclusion (OR = 1.89; *p* = 0.039) and tuberculate morphology (OR = 2.93; *p* = 0.021) as the strongest independent predictors of impaction, alongside associations with younger age (<13 years) (OR = 3.12; *p* = 0.003) and male gender (OR = 1.78; *p* = 0.046). **Conclusions:** This multicentric study provides updated Romanian data and highlights novel risk factors and diagnostic selection guidelines that may support individualized treatment planning. Angle Class III malocclusion is a novel and critical independent predictor of supernumerary tooth impaction, alongside tuberculate morphology. This finding strengthens the rationale for utilizing CBCT specifically in Class III patients with ST to pre-emptively manage complex impactions and associated pathology.

## 1. Introduction

Supernumerary teeth (ST), defined as a developmental anomaly representing an excess of the normal dental arch count—32 in the permanent and 20 in the primary dentition—constitute a complex clinical condition that can significantly affect both the esthetic and functional integrity of the dento-maxillary system [1]. These additional teeth, which display considerable variation in shape, size, position, and eruption pattern, are a subject of ongoing research focusing on their classification, epidemiological distribution, and multifactorial etiopathogenesis [2,3].

The presence of supernumerary teeth (Figure 1) is not a morphological curiosity but a potential source of diverse pathological complications [1,2,3,4].

The most frequently encountered issues include delayed eruption or impaction, displacement, and malformation of adjacent permanent teeth [4,5]. In more severe cases, ST may induce root resorption, cystic formations, or ectopic migrations into the nasal cavity or maxillary sinus [6,7,8,9,10]. Furthermore, these anomalies can interfere with alveolar bone grafting, orthodontic alignment, or implant placement, and in rare instances, have been associated with neurological disturbances such as neuralgia or paresthesia [11,12,13]. Therefore, early detection and accurate diagnosis are crucial to mitigate secondary complications [14,15,16,17].

### 1.1. Etiopathogeny, Epidemiology, and Clinical Relevance

The etiopathogenesis of supernumerary teeth remains elusive, although most authors agree that their occurrence reflects a developmental disturbance during organogenesis and dental morphogenesis [2,18,19,20]. Several hypotheses have been proposed to elucidate this complex process. One of the earliest is the tooth bud division theory, which suggests that a single tooth germ divides partially or completely, resulting in two distinct dental structures [2,18,19]. Another explanation, known as the third dentition theory, proposes that supernumerary teeth arise from the reactivation of dental lamina remnants during a hypothetical third cycle of tooth development, following the primary and permanent dentitions [2,20,21].

A further perspective considers the proliferation of epithelial remnants, wherein residual elements of the dental lamina or Hertwig’s epithelial root sheath, persisting in the jaws after the normal tooth development process, may undergo abnormal proliferation and differentiation under the influence of local or systemic stimuli, giving rise to additional tooth structures [2,19].

The dental lamina hyperactivity theory has gained the widest acceptance in contemporary literature. This hypothesis attributes the formation of supernumerary teeth to localized and independent hyperactivity of the dental lamina, which leads to the development of supplementary epithelial buds during embryogenesis [19,20,22,23]. Such hyperactivity may be triggered by vascular, histochemical, or developing structural environment in the craniofacial region, affecting the delicate regulatory mechanisms that control tooth initiation and morphogenesis [21,22].

Overall, these theories indicate that the genesis of supernumerary teeth is multifactorial, arising from an interplay between genetic susceptibility, local developmental disturbances, and environmental influences that alter the normal morphogenetic potential of the dental lamina [3,24].

The increased frequency of ST in the maxillary incisor region (Figure 2) may be linked to intricate tissue rearrangements and morphogenetic activity in this area [2,17,20,25]. Nevertheless, no single theory fully accounts for the wide diversity of clinical presentations, suggesting that the etiology of ST is multifactorial, shaped by a complex interplay between genetic predisposition and environmental influences [3,24].

Genetic studies have revealed familial clustering of cases and associations with syndromic conditions such as Gardner’s syndrome and cleidocranial dysplasia [25,26]. Brook’s unified etiological model [2,26] postulates that supernumerary teeth result from genetically determined susceptibility, modulated by environmental and epigenetic factors. This framework reflects the modern understanding of dental developmental anomalies as polygenic and multifactorial disorders rather than isolated occurrences.

### 1.2. Epidemiological Overview

Epidemiological studies estimate the prevalence of supernumerary teeth to be 0.1% to 3.8% in the general population, with a distinct male predominance (approximately 2:1) and a higher incidence in the permanent dentition [17,27]. The anterior maxilla represents the most common site of occurrence, particularly the mesiodens type. Approximately 75% of supernumerary teeth are impacted and asymptomatic, often discovered incidentally during routine radiographic assessments [2]. To provide a clearer understanding of the occurrence and clinical behavior of supernumerary teeth, Table 1 compiles essential epidemiological and clinical data reported in the literature.

The clinical impact of ST varies with location, morphology, and eruption status [29]. While some remain clinically silent, others contribute to malocclusion, delayed eruption of permanent successors, or diastemas, and may cause esthetic and functional disturbances that require multidisciplinary management [30,31,32].

### 1.3. Diagnostic Essentials

Accurate diagnosis and precise localization are indispensable for successful treatment planning and prevention of secondary complications [2,18]. Traditional two-dimensional radiographic methods [33], such as panoramic (orthopantomogram) and periapical radiographs (Figure 3), remain valuable as initial diagnostic tools; however, they often fail to provide sufficient information regarding the three-dimensional relationship between the ST and adjacent anatomical structures [33,34].

Cone-beam computed tomography (CBCT) addresses these limitations and has become a key diagnostic adjunct in contemporary diagnostic protocols [35]. CBCT allows high-resolution, three-dimensional imaging from a single scan, offering detailed visualization of supernumerary tooth morphology, spatial orientation (Figure 4), and proximity to vital structures such as the nasal floor, maxillary sinus, and mandibular canal [35,36].

CBCT was performed exclusively when the following are clinically indicated: ambiguous OPG, suspected lateral impaction, complex morphology, or proximity to vital structures. All examinations followed ALARA/ALADAIP principles, using the minimum FOV and voxel size required for diagnostic reliability. This study provides real-world evidence for selective CBCT indications based on morphology and location. Moreover, CBCT has been shown to enhance the detection of associated pathologies, such as root resorption and cyst formation, with greater precision than conventional radiography [35,36]. Nevertheless, given the higher radiation exposure, its use should be carefully justified based on a risk-benefit assessment for each case [37,38].

AI systems that utilize CBCT scans can accurately interpret the images and recommend the most suitable surgical strategy for managing supernumerary teeth. This technology makes the operation easier and faster to perform [15]. Diagnostic performance metrics comparing OPG with CBCT for the detection of impacted and laterally positioned supernumerary teeth are provided in Appendix A.

### 1.4. Management Strategies and Current Controversies

The therapeutic management of supernumerary teeth generally follows two principal approaches, extraction or non-extraction (clinical monitoring), with the choice largely determined by the tooth’s position, its relationship to adjacent anatomical structures, and the presence or risk of associated pathology [39]. In cases where supernumerary teeth interfere with eruption, alignment, or occlusion, surgical removal is typically indicated. Asymptomatic and non-disruptive teeth may be observed periodically through clinical and radiographic follow-up [39,40].

A persistent point of debate within the literature concerns the optimal timing of surgical intervention [41,42,43]. The late approach advocates postponing extraction until after the apical closure of the permanent incisors, generally around the age of ten years. This strategy minimizes the risk of damaging developing roots and surrounding periodontal structures, thereby preserving long-term dental integrity. In contrast, the early approach supports immediate removal upon diagnosis, particularly when the supernumerary tooth exhibits an oblique orientation, distorted morphology, or a position that makes spontaneous eruption improbable [18,41,42,43].

Ultimately, the decision regarding timing must balance the risks of surgical trauma against the potential for eruption disturbances, the individual’s age, tooth development stage, and overall orthodontic and esthetic prognosis [42,43]. Some clinicians recommend avoiding intervention in asymptomatic cases, stressing the value of long-term monitoring to prevent unnecessary surgical risks [44,45]. This ongoing debate highlights the importance of employing evidence-based, individualized management strategies that rely on comprehensive diagnostic evaluations and collaboration across disciplines.

Despite numerous international reports, data from Eastern Europe remain limited and lack standardized radiographic protocols [4]. Crucially, few prior studies jointly evaluate morphology, skeletal class (Angle classification), and age as predictors of impaction within a multivariable statistical model, and even fewer provide comparative, evidence-based criteria for OPG and CBCT indications [5]. Our study was designed to address these distinct gaps by providing contemporary Romanian data and novel multivariate predictors of impaction [6].

## 2. Materials and Methods

This multicentric, observational, analytical study was conducted between January 2020 and March 2025 across three Romanian academic centers: the University Dental Clinic, George Emil Palade University of Medicine, Pharmacy, Science, and Technology of Târgu Mureș; the Periodontology Department, Faculty of Medicine, Lucian Blaga University of Sibiu; and the Department of Dentistry, Faculty of Medicine, Dimitrie Cantemir University of Târgu Mureș. A retrospective review of consecutive clinical and radiographic records from these centers identified 153 eligible patients with one or more clinically and radiographically confirmed supernumerary teeth. Patient accrual by center was as follows: George Emil Palade University (Târgu Mureș): *n* = 78; Lucian Blaga University (Sibiu): *n* = 45; Dimitrie Cantemir University (Târgu Mureș): *n* = 30. Consecutive sampling was used to minimize selection bias. The study adhered to the STROBE reporting guidelines for observational research. Written informed consent was obtained from all participants or legal guardians.

### 2.1. Inclusion and Exclusion Criteria

Inclusion criteria consisted of patients presenting with one or more supernumerary teeth in the mixed or permanent dentition, along with complete diagnostic records including an orthopantomogram (OPG) and/or cone-beam computed tomography (CBCT). Exclusion criteria included syndromic disorders (e.g., cleidocranial dysplasia, Gardner’s syndrome), craniofacial trauma, prior orthodontic or surgical interventions, or incomplete imaging documentation.

### 2.2. Clinical and Radiographic Examination

Clinical examination included assessment of eruption status, dental crowding, malposition, and associated cystic or resorptive changes. Radiological evaluation followed a standardized diagnostic protocol:OPG for screening and global assessment (Digital panoramic radiography obtained using a ProScan PanoSight 2000 unit from MedTech Imaging Solutions GmbH, Vienna, Austria, processed with the dedicated PanoSight Viewer software, version 3.4.1. for screening and global assessment).Periapical and occlusal radiographs to refine localization (IntraFocus X-Ray i80 intraoral imaging system of DentalVision SRL, Cluj-Napoca, Romania, analyzed with DentalVision Capture software, version 2.7.0.).CBCT examinations (OrthoCube 3D Max device of Imaging Dynamics Europe Sp. z o.o., Warsaw, Poland, and CubeScan 3D Studio software, version 5.2.3) were reserved exclusively for complex or ambiguous cases, using a voxel resolution of 0.2–0.3 mm, an 8 × 8 cm field of view, and exposure parameters of 90 kVp and 10 mA. Indications included unclear findings on panoramic radiography, suspected lateral impactions, complex root or crown morphology, and situations involving proximity to vital anatomical structures. In all cases, the ALARA/ALADAIP principles were strictly followed, ensuring the lowest voxel size necessary for reliable diagnosis were used. As such, this study provides real-world evidence supporting selective, morphology- and location-based indications for CBCT imaging.

### 2.3. Classification Criteria

Supernumerary teeth were categorized according to the following:-Morphology: conical, tuberculate, supplemental, or odontomatous.-Topography: mesiodens, paramolar, distomolar, or supernumerary premolar.-Eruption status: erupted, impacted, or included.

Definitions:-Erupted: Visible in the oral cavity and in functional occlusion.-Impacted: Completely unerupted and entirely within bone, with no mucosal penetration.-Included: Unerupted tooth partially covered by mucosa or soft tissue.-Erupted and impacted (lateral): Unerupted tooth partially covered by mucosa or soft tissue, typically with ectopic lateral displacement [10]. This classification was necessary to accurately describe a distinct subgroup of partial-eruption cases with angular deviation noted in the cohort. Both examiners have used these definitions for case descriptions.

Malocclusion was classified using Angle’s classification (Class I, II/1, II/2, III) to assess potential occlusal associations.

### 2.4. Bias Mitigation

To mitigate potential sources of bias inherent in this observational study design, several steps were taken:-Selection Bias: Sampling for all eligible patients seen between January 2020 and March 2025 aimed to minimize selection bias by preventing the deliberate exclusion or inclusion of patients based on their complication severity.-Information Bias (Measurement Bias): To ensure high reliability of radiological data, two calibrated examiners (oral radiologists) independently analyzed all images. The subsequent calculation and verification of excellent inter-observer agreement (kappa = 0.91) standardized the radiographic interpretation, limiting measurement bias.-Confounding Bias: The study controlled for several known potential confounders of impaction, including age, gender, and malocclusion, by including these variables in the Binary Logistic Regression model (as detailed in Section 2.5).

Because of the study’s retrospective and multicenter design, a priori power analysis was not conducted. Instead, the final sample size of 153 consecutive patients represented the maximum available population meeting all inclusion criteria within the five-year study period (2020–2025). Based on the final logistic regression model, which identified four significant independent predictors of impaction using 153 cases, the sample size was deemed sufficient to achieve appropriate power to detect the reported effects (OR = 1.78 to 3.12) with a *p* < 0.05 and a confidence level of 95%. Inter-observer agreement was excellent (Cohen’s κ = 0.91). Discrepancies (<5%) were resolved by consensus review; persistent disagreements were adjudicated by a senior radiologist.

### 2.5. Ethics and Governance

This retrospective multicentric study was reviewed and approved by the institutional ethics committees of the participating centers. Ethical approvals were granted by the following parties:-The Ethics Committee of the George Emil Palade University of Medicine, Pharmacy, Science, and Technology of Târgu Mureș, by Decision No. 1885 of 12 October 2022;-Scientific Research Ethics Committee of the Lucian Blaga University of Sibiu through ethical approval number 33/15 March 2023;-The Scientific Research Ethics Committee of Dimitrie Cantemir University of Târgu Mureș, Romania by Approval No. 131 of 19 July 2021.

Written informed consent was obtained from all adult participants and from parents/legal guardians of minors. All study procedures were conducted in accordance with the Declaration of Helsinki and national regulations on research on human subjects.

All transferred datasets were de-identified before central analysis and stored on a password-protected institutional server. Requests for de-identified data may be considered by the corresponding authors following institutional procedures and approval by the relevant ethics committees.

### 2.6. Statistical Analysis

Variables included in the multivariable logistic regression model were selected a priori based on clinical relevance (age group, sex, morphology, anatomical location, and Angle classification). Age was evaluated both as a dichotomized variable (<13 vs. ≥13 years, reflecting dentition stage) and, in sensitivity analyses, as a continuous predictor, and using alternative cut-offs (10 and 12 years). The logistic model included 70 impaction events across six predictors, yielding an events-per-variable ratio (EPV) of 11.6, consistent with accepted standards for model stability.

Multicollinearity was assessed using Variance Inflation Factors (VIF); all predictors remained below the threshold of 2.5. Model calibration was evaluated using the Hosmer–Lemeshow goodness-of-fit test, and model discrimination was quantified using the area under the ROC curve (AUC). Missing data were <2% and were managed using complete-case analysis; multiple imputation (m = 5) was applied as a robustness check, yielding consistent estimates.

Results are reported as adjusted odds ratios (OR) with 95% confidence intervals (CI). Statistical significance was set at *p* < 0.05. The complete model coefficients, diagnostic statistics, and sensitivity analyses are provided in Table 1. Model discrimination was further evaluated using ROC analysis, with the ROC curve presented in Appendix A.

## 3. Results

### 3.1. Demographic Characteristics

The study population consisted of 153 patients (91 males, 62 females; mean age 14.8 ± 6.2 years). Patient recruitment by center was George Emil Palade University, Târgu Mureș—78 patients (51.0%); Lucian Blaga University, Sibiu—45 patients (29.4%); Dimitrie Cantemir University, Târgu Mureș—30 patients (19.6%). Most participants (47%) were aged 6–12, and 67.3% resided in urban areas. A total of 185 supernumerary teeth were identified (mean 1.21 ± 0.56 per patient), with single occurrences in 83.0%, double in 13.1%, and multiple (>2) in 3.9% of cases.

Table 2 shows the distribution of patients by gender and dental class according to Angle’s classification.

### 3.2. Distribution by Gender and Tooth Position

Impaction patterns differed significantly by gender. Males (56%) exhibited more frequent impactions, particularly in anterior regions. As presented in Table 3, most erupted teeth occurred in the lateral zones, whereas combined frontal and lateral impactions were more common in males.

### 3.3. Age-Related Distribution

Younger patients (<13 years) exhibited significantly higher impaction rates (OR = 3.12, *p* = 0.003). The detailed age-group distribution is presented in Table 4.

### 3.4. Morphological and Topographical Features

Morphologically, conical teeth were most common (48.6%), followed by tuberculate (27.0%), supplemental (18.4%), and odontomatous (6.0%) forms. Topographically, mesiodens accounted for over half of the total (56.2%), followed by paramolar (16.2%), distomolar (14.1%), and supernumerary premolar (13.5%) variants (Table 5). Tuberculate teeth were impacted in 72% (36/50), compared with 41% (37/90) of conical teeth.

### 3.5. Clinical Complications

Clinical complications were observed in 40.5% of patients. The most frequent were delayed eruption (19.6%), followed by malposition or rotation (12.4%), cystic formation (4.6%), and root resorption (3.9%). Details are presented in Table 6.

### 3.6. Radiographic Findings

OPG successfully detected 92.4% of supernumerary teeth, whereas CBCT confirmed all cases and identified 11 root resorptions and 9 vestibulo-palatal displacements missed by the OPG. The diagnostic concordance between the two imaging modalities was high (κ = 0.89), supporting CBCT’s superior sensitivity for spatial localization and pathological assessment. As shown in Appendix A, OPG demonstrated high specificity (93%) and good sensitivity (80%) relative to CBCT, with most false negatives occurring in laterally displaced or tuberculate teeth.

### 3.7. Predictors of Impaction

Binary logistic regression identified four factors as independent predictors of impaction (Table 7): younger age (<13 years), male gender, tuberculate morphology, and Angle Class III malocclusion.

Model diagnostics showed good calibration (Hosmer–Lemeshow *p* = 0.48) and acceptable discrimination (AUC = 0.79). The ROC curve illustrating model discrimination (AUC = 0.79) is provided in Appendix A. All predictors demonstrated low multicollinearity (VIF range 1.06–1.87). Sensitivity analyses using age as a continuous predictor and cut-offs of 10 and 12 years produced results consistent with the primary model. Complete regression coefficients, diagnostic statistics, and all sensitivity analyses are presented in Appendix A.

### 3.8. Correlations

Spearman’s correlation revealed a moderate positive relationship between age and the number of supernumerary teeth (ρ = 0.33, *p* < 0.001), indicating a gradual increase in multiplicity with advancing age, likely due to progressive detection over time.

The study demonstrates a clear male predominance, high frequency of anterior mesiodens impactions, and strong age-dependent and morphology-dependent patterns of pathology. CBCT provided superior diagnostic accuracy, and impaction was significantly predicted by early age, male sex, tuberculate form, and Class III occlusal pattern.

## 4. Discussion

The most significant and novel finding of this study is the identification of Angle Class III malocclusion as an independent predictor of supernumerary tooth impaction (OR = 1.89; *p* = 0.039). This association is biologically plausible. Angle Class III malocclusion often involves an enlarged mandible (prognathism) and/or a restricted maxilla, leading to an overall reduction in the available space in the anterior maxillary region [34]. This skeletal discrepancy likely contributes to a more horizontal growth vector and subsequent impaction or ectopic eruption pathway for supernumerary teeth. Few studies have explicitly examined this relationship, with most focusing solely on eruption space. Our finding suggests that the underlying skeletal pattern, not just localized space deficiency, is a systemic contributing factor to impaction risk.

This multicenter study underscores the clinical value of systematically characterizing the epidemiological, clinical, and imaging features of supernumerary teeth within a Romanian population. By integrating morphological assessment, occlusal analysis, and detailed radiological evaluation, the investigation highlights how these factors interact to influence the likelihood of impaction. Its findings not only confirm but also broaden current knowledge on the behavior and clinical implications of supernumerary teeth, emphasizing the need for individualized, imaging-based diagnostic and treatment pathways. This retrospective study used consecutive data from three academic referral centers; therefore, the sample benefits from standardized imaging protocols, and absolute prevalence estimates may reflect referral bias toward more complex cases.

Importantly, the study introduces three notable contributions to the field: (1) the first Romanian multicenter cohort evaluated with a standardized OPG + CBCT protocol, establishing a consistent framework for cross-center comparison, (2) identification of tuberculate morphology and Class III malocclusion as independent predictors of impaction, based on robust multivariable modeling, and (3) development of pragmatic criteria for selective CBCT indication, allowing clinicians to tailor imaging choices to patient-specific risk profiles.

Together, these advances move beyond traditional descriptive reporting, offering a risk-stratified approach that directly informs everyday clinical decision-making and supports more precise, evidence-guided management of patients with supernumerary teeth.

The male predominance (59.5%) observed aligns with the widely reported gender disparity, confirming previous data from European and Asian populations, where male-to-female ratios range from 1.4:1 to 3.2:1 [18,35,46,47,48]. This trend may reflect the influence of sex-linked genetic factors or differential expression of odontogenic regulatory genes such as *WNT10A* or *RUNX2*, known to participate in dental lamina signaling pathways [49]. The predominance of detection between 6 and 12 years further emphasizes the role of panoramic screening during early mixed dentition, when most supernumerary formations become radiographically visible and can be intercepted before causing eruption disturbances [50].

Morphologically, the conical form was the most prevalent (48.6%), consistent with earlier studies by Davidson et al. (2025) and Rajab and Hamdan (2002), which reported similar frequencies of 45–55% [2,51]. The second most common type, tuberculate teeth (27%), demonstrated a distinct pathological profile, showing stronger associations with delayed eruption and cystic transformation [2,15]. This is biologically plausible, as tuberculate teeth possess irregular morphology and multicuspid crowns that impede the normal eruption pathway, predisposing to impaction and pericoronal cyst development [52]. Supplemental and odontomatous forms in a smaller proportion support the theory that hyperactivity of the dental lamina leads to variable degrees of morphological differentiation [20,21,22,23].

Topographically, mesiodens accounted for 56.2% of all supernumerary teeth, confirming that the anterior maxilla is the predominant site of occurrence [1,2,28]. This localization correlates with the area of greatest embryonic epithelial complexity and supports the dental lamina hyperactivity theory as the prevailing etiopathogenic model [1,2,21,22,23,28]. The clustering of supernumerary teeth in this region may also be influenced by mechanical and spatial constraints during incisor eruption, explaining the higher frequency of impactions in anterior sectors [1,52,53,54].

From an orthodontic standpoint, the distribution of Angle classes provides a novel dimension to understanding the skeletal context of supernumerary teeth. The significant association between Class III malocclusion and anterior impactions (*p* = 0.008) represents an original finding. Similar tendencies have been noted in Korean and Turkish cohorts, where concave facial growth patterns and anterior cross-bites were correlated with restricted eruption spaces [55,56,57]. This relationship may be mediated by anteroposterior maxillary deficiency, altering the eruption trajectory of incisors and promoting mesiodens impaction.

The logistic regression model strengthened these associations, identifying younger age (<13 years), male gender, tuberculate morphology, and Class III malocclusion as independent predictors of impaction. Younger patients showed a 3.1-fold higher risk, reflecting developmental timing, since many permanent successors are still unerupted, and increased detectability in early radiographic surveys. Male gender nearly doubled the odds of impaction (OR = 1.78), paralleling reports from Ferrés-Padró et al. (2013) and Patil et al. (2020) [58,59], which attributed this tendency to sexual dimorphism in craniofacial growth and tooth bud spacing. The finding that tuberculate morphology increases cystic potential nearly three-fold (OR = 2.93) supports the notion that crown form directly influences pathological sequelae.

The positive correlation between age and the number of supernumerary teeth (ρ = 0.33, *p* < 0.001) can be interpreted as a detection bias but also indicates that multiple ST, especially distomolars and premolars, mineralize later and thus appear progressively in older patients. Comparable age-related trends were reported by Anthonappa et al. (2012), reinforcing the need for long-term radiographic follow-up in patients with known mesiodens, as additional teeth may develop subsequently [33].

Radiographically, this study confirmed the diagnostic superiority of CBCT over orthopantomography. While OPG detected 92.4% of ST, CBCT achieved full detection and revealed additional findings—11 unrecognized root resorptions and 9 vestibulo-palatal displacements. The high inter-modality concordance (κ = 0.89) supports the reliability of panoramic screening for preliminary assessment, validating CBCT as the gold standard for surgical planning, particularly for teeth adjacent to vital structures or in complex morphologies [60,61]. These observations echo the results of Primosch (1981) and recent analyses by Gündüz et al. (2022), emphasizing CBCT’s critical role in minimizing iatrogenic risks during extraction [62,63]. However, consistent with ALARA principles, CBCT should be reserved for cases where conventional imaging is inconclusive or where three-dimensional spatial orientation dictates treatment choice [64,65].

Clinically, 40.5% of patients presented complications, with delayed eruption (19.6%) as the most frequent, followed by malposition (12.4%), cyst formation (4.6%), and root resorption (3.9%). These frequencies are comparable to the 30–40% complication rates described by Fardi et al. (2011) and Kara et al. (2012) [66,67]. The link between tuberculate morphology and cystic change observed in our series mirrors the findings of He et al. (2023), who reported cystic transformation in 9–11% of cases, increasing with age [35]. Thus, early diagnosis prevents impaction and mitigates the risk of secondary pathologies that complicate orthodontic management. Our findings demonstrate that tuberculate morphology substantially increases the odds of impaction, likely due to altered crown shape and eruption path obstruction. Additionally, the association with Class III malocclusion may reflect midline positional shifts and altered anterior eruption trajectories. These results emphasize the need for earlier surveillance and individualized treatment planning in patients with skeletal discrepancies or complex morphology.

The strong relationship between morphology, skeletal class, and impaction risk provides clinically actionable guidance for radiographic selection. Specifically, OPG remains reliable for frontal conical ST, whereas tuberculate or laterally displaced teeth justify CBCT evaluation. This selective approach balances diagnostic accuracy and radiation minimization.

Overall, the findings support early radiographic screening, ideally between 6 and 8 years, when eruption disturbances can be anticipated and minimally invasive extraction can be performed before root maturation of permanent successors. However, the debate regarding early versus delayed surgical removal remains relevant. Our data suggest early intervention for obliquely positioned or tuberculate forms, and delayed extraction after apical closure may be justified for conical mesiodens without associated complications. Therefore, a balanced, patient-specific approach remains fundamental.

Despite employing rigorous consecutive sampling and high inter-rater reliability, the study acknowledges several limitations inherent to its multicentric, observational design. This study is retrospective and based on university referral centers, which may enrich the sample with more symptomatic or complex cases. Therefore, absolute prevalence rates should not be generalized to the broader population; however, relative associations (adjusted predictors) are less susceptible to referral bias. Despite robust multivariable modeling, prospective longitudinal studies are warranted to confirm causal pathways and eruption outcomes. Additionally, Romanian patients may differ from other populations; therefore, cross-national studies could further validate these associations. Future research in this field is poised to move beyond isolated observations and toward a truly integrative understanding of how supernumerary teeth emerge, evolve, and affect dental development. Longitudinal follow-up will be essential—not only to compare eruption outcomes between early and delayed interventions, but also to monitor recurrence patterns and the possible appearance of additional supernumerary teeth over time. As these clinical pathways are traced, parallel molecular investigations may illuminate the underlying mechanisms: genomic and epigenetic analyses, including targeted mapping of WNT10A, RUNX2, and AXIN2 polymorphisms, could reveal how subtle shifts in gene regulation trigger dental lamina hyperactivity.

Equally important is the expansion of study populations. Broadening ethnic and age representation will strengthen the applicability of future findings and capture variations that smaller, homogeneous samples might miss. Complementing this diversity, advanced imaging technologies—particularly 3D morphometric analyses—offer the opportunity to quantify how crown shape and spatial relationships influence eruption behavior.

Alfailany et al. evaluated the diagnostic accuracy of CBCT and two-dimensional imaging for impacted canines and found improved localization and assessment using CBCT [68]. Hajeer et al. [69] described a structured CBCT vs. 2D interpretation panel method and reported inter-observer reliability metrics supporting standardized calibration procedures. The next frontier may lie at the intersection of clinical imaging and computational intelligence. AI-assisted radiographic evaluation, especially models that fuse CBCT and OPG features, holds promise for elevating diagnostic accuracy, reducing inter-observer variability, and enabling scalable assessment in large cohorts. In uniting biological inquiry, imaging innovation, and computational precision, forthcoming research can redefine both the understanding and management of supernumerary teeth.

Supernumerary teeth represent a multifactorial developmental anomaly with diverse clinical presentations and potential complications. Their management demands a careful balance between timely intervention and conservative monitoring, guided by accurate imaging and individualized clinical judgment. Spanning genetic, environmental, and developmental factors, the complexity of their etiology underscores the importance of continued research to refine diagnostic methods and establish standardized treatment protocols. Ultimately, successful management of ST relies on an integrated approach that unites the expertise of orthodontists, oral surgeons, and radiologists to ensure optimal functional and esthetic outcomes.

This study successfully characterizes the morphological and epidemiological patterns of supernumerary teeth, identifies risk factors for impaction, and evaluates diagnostic tools.

## 5. Conclusions

This multicentric Romanian study contributes novel insights by identifying Class III skeletal pattern and tuberculate morphology as independent predictors of impaction and by providing evidence-based criteria for selective CBCT use. These findings support individualized surveillance and diagnostic pathways for patients with supernumerary teeth and highlight the value of integrating morphological and skeletal assessments in clinical decision-making.

## Figures and Tables

**Figure 1 diagnostics-15-03019-f001:**
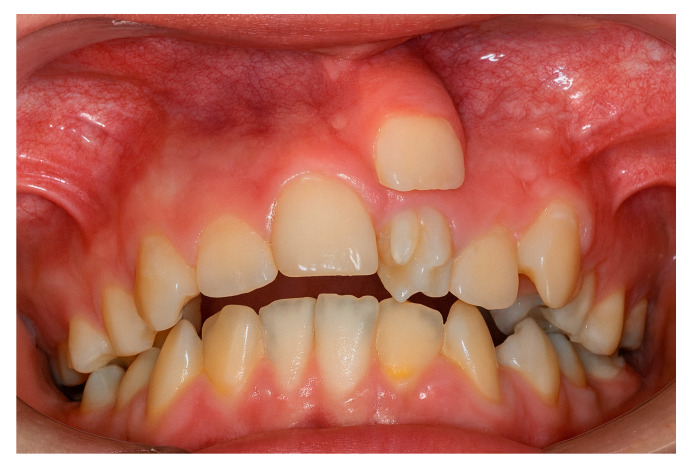
Supernumerary teeth in the upper left quadrant.

**Figure 2 diagnostics-15-03019-f002:**
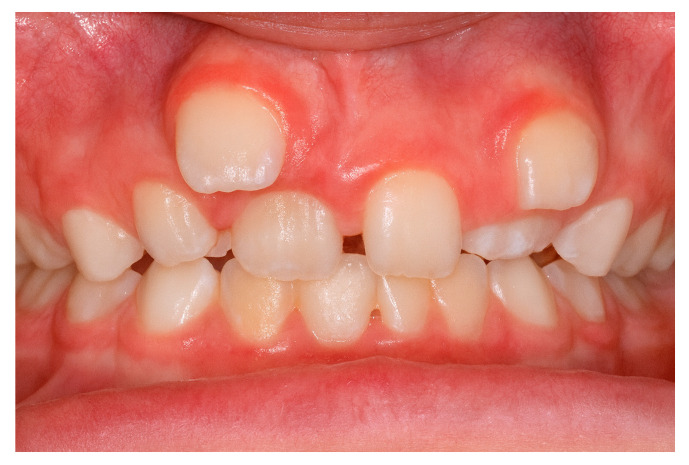
Supernumerary teeth located in the maxillary incisor area.

**Figure 3 diagnostics-15-03019-f003:**
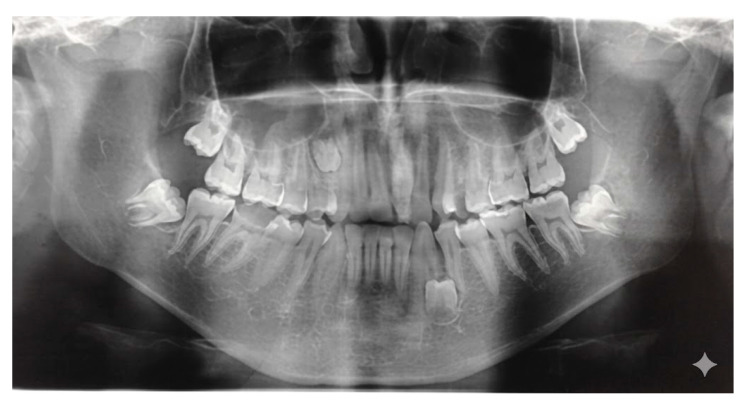
Orthopantomogram showing a supernumerary tooth located between the left mandibular canine (tooth 3.3) and the left mandibular first premolar (tooth 3.4).

**Figure 4 diagnostics-15-03019-f004:**
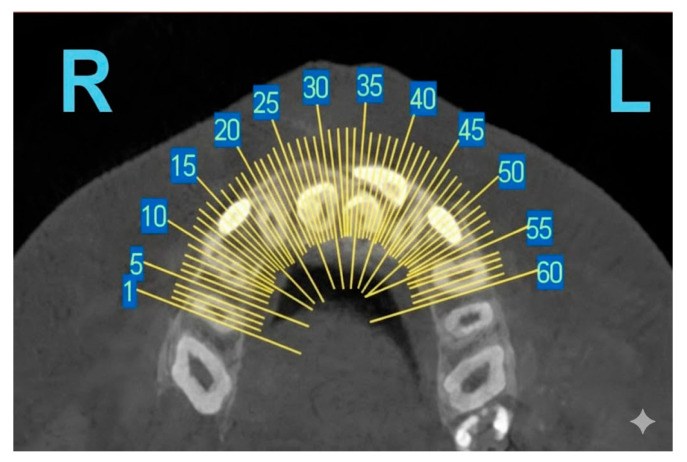
Representative CBCT slice.

**Table 1 diagnostics-15-03019-t001:** Epidemiological and Clinical Profile of Supernumerary Teeth.

Key Epidemiological and Clinical Data	Value/Observation	Source
Prevalence in the General Population	0.1–3.8%	[27]
Gender Predominance	Male (approx. 2:1 ratio)	[27]
Most Frequent Location	Anterior Maxilla (Mesiodens)	[2,28]
Impaction/Asymptomatic Rate	~75%	[2,28]
Reported Complication Frequency	21.6–30%	[20]
Cystic Formation Frequency	9–11%	[20,25]

**Table 2 diagnostics-15-03019-t002:** Distribution of Patients by Gender and Dental Class (Angle).

Gender	Class I	Class II/1	Class II/2	Class III	Total
Male	28	21	10	32	91
Female	14	12	8	28	62
Total (*n* = 153)	42 (27.5%)	33 (21.6%)	18 (11.8%)	60 (39.2%)	100%

Note: A significant association was observed between Angle class and supernumerary tooth location (chi^2^ = 11.78, *p* = 0.008). Percentages refer to the number of patients (*n* = 153).

**Table 3 diagnostics-15-03019-t003:** Distribution of Patients by Gender and Supernumerary Tooth Position.

Gender	Erupted	Erupted and Impacted (Lateral Zone)	Impacted (Frontal Zone)	Impacted (Frontal and Lateral)	Impacted (Lateral)	Unspecified	Total
Male	38	12	6	19	11	5	91
Female	25	10	3	12	8	4	62
Total (*n* = 153)	63 (41.2%)	22 (14.4%)	9 (5.9%)	31 (20.3%)	19 (12.4%)	9 (5.9%)	100%

Note: Percentages refer to the number of patients (*n* = 153).

**Table 4 diagnostics-15-03019-t004:** Distribution of Patients by Age Group and Supernumerary Tooth Position.

Age Group (Years)	Erupted	Erupted and Impacted (Lateral)	Impacted (Frontal)	Impacted (Frontal and Lateral)	Impacted (Lateral)	Unspecified	Total
6–12	30	14	4	12	8	4	72
13–18	20	6	3	10	6	3	48
19–45	13	2	2	9	5	2	33
Total (*n*= 153)	63 (41.2%)	22 (14.4%)	9 (5.9%)	31 (20.3%)	19 (12.4%)	9 (5.9%)	100%

Note: Percentages refer to the number of patients (*n* = 153).

**Table 5 diagnostics-15-03019-t005:** Morphological and Topographical Distribution of Supernumerary Teeth (*n* = 185).

Variable	Category	Frequency (*n*)	Percentage (%)
Morphology	Conical	90	48.6
	Tuberculate	50	27.0
	Supplemental	34	18.4
	Odontomatous	11	6.0
Location	Mesiodens	104	56.2
	Paramolar	30	16.2
	Distomolar	26	14.1
	Supernumerary Premolar	25	13.5

Note: Percentages refer to the number of supernumerary teeth (*n* = 185).

**Table 6 diagnostics-15-03019-t006:** Complications Associated with Supernumerary Teeth.

Complication	Number of Patients (*n*)	Percentage (%)
Delayed eruption of adjacent teeth	30	19.6
Malposition/rotation	19	12.4
Cystic formation	7	4.6
Root resorption	6	3.9
No complications	91	59.5

Note: Percentages refer to the number of patients (*n* = 153).

**Table 7 diagnostics-15-03019-t007:** Logistic Regression Predictors of Supernumerary Tooth Impaction.

Variable	Odds Ratio (OR)	95% CI	*p*-Value
Age < 13 years	3.12	1.47–6.64	0.003
Male gender	1.78	1.01–3.15	0.046
Tuberculate morphology	2.93	1.17–5.72	0.021
Class III malocclusion	1.89	1.03–3.02	0.039

## Data Availability

The original contributions presented in this study are included in the article/Appendix A. Further inquiries can be directed to the corresponding authors.

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
