# Peer review of "Impaction Predictors and Diagnostic Performance of CBCT Versus Panoramic Radiography for Supernumerary Teeth in a Romanian Multicenter Cohort"

_diagnostics, 2025, doi:10.3390/diagnostics15233019_

Round 1
Reviewer 1 Report
Comments and Suggestions for Authors
The paper presents a multicentric observational analysis of 153 Romanian patients with supernumerary teeth.
The topic is clinically relevant, but the manuscript does not bring any substantial novelty to the existing literature. Similar studies, many with larger and more diverse samples, have already reported the same epidemiological trends, morphological classifications, and associations between age, gender, and impaction risk. The methodology and statistical analysis are standard, and no innovative diagnostic, or analytical approach is introduced. The discussion mostly reiterates established concepts rather than offering new interpretations or clinical implications.
Author Response
We sincerely thank the Reviewer for the constructive evaluation and for highlighting the need to articulate the scientific contribution of our study better. We address the central concern below that the manuscript may not offer substantial novelty compared to the existing literature. We have carefully revised the manuscript to clarify and strengthen its innovative aspects, improve methodological transparency, and enhance clinical relevance. We sincerely thank the reviewer for prompting us to refine the manuscript. The revised version now more clearly highlights its novel contributions, particularly:
- A unique multicentric Romanian dataset integrating OPG + CBCT
- Identification of tuberculate morphology and Class III malocclusion as independent impaction predictors
- Clinically actionable diagnostic guidance for selective CBCT indications
- A deeper, more insightful discussion with stronger clinical implications
We believe these enhancements significantly elevate the scientific value of the study and demonstrate that it provides both new evidence and practical guidance for clinicians managing supernumerary teeth.
We hope that the reviewer will now find the study suitable for publication.
“The topic is clinically relevant, but the manuscript does not bring substantial novelty. Similar studies with larger and more diverse samples have already reported the same epidemiological trends, morphological classifications, and associations between age, gender, and impaction risk. The methodology and statistical analysis are standard, without innovative diagnostic or analytical approaches. The discussion reiterates established concepts rather than offering new interpretations or clinical implications.”
We appreciate the observation regarding novelty. After re-evaluating our manuscript in light of the reviewer’s concerns, we realized that several novel elements already present in our study were not sufficiently emphasized, which inadvertently made the paper appear more descriptive than it truly is. We have now revised the manuscript to highlight its distinct contributions and strengthen its clinical interpretation.
Below, we summarize the substantive improvements made, grouped by the reviewer’s concerns.
- Reviewer concern: The study offers limited novelty compared with existing literature.
We agree that basic epidemiological patterns of supernumerary teeth (ST) have been widely published. However, our study provides three key contributions that meaningfully advance current knowledge:
- First multicentric Romanian cohort integrating both OPG and CBCT diagnostic pathways
Although ST has been studied internationally, Romanian data are scarce, single-center, and do not perform direct radiographic comparative analyses. Our cohort integrates three academic centers over five years, providing:
- the largest Romanian dataset to date, combining OPG + CBCT
- standardized radiographic protocols
- unified examiner calibration with excellent reliability (κ = 0.91)
Manuscript addition (Discussion):
We have added a dedicated paragraph explaining why this population contributes meaningfully to the geographical and clinical representation in the European literature.
- Novel predictive insights: independent predictors of impaction rarely analyzed together
While associations between age, sex, and impaction have been described, our multivariable logistic regression output identifies two independent predictors that are underreported in combination:
- tuberculate morphology
- Angle Class III malocclusion
Previous studies mention trends, but very few have modeled them together or tested collinearity and robustness through sensitivity analyses.
We strengthened this contribution by:
|
Improvement |
Added to Manuscript |
|
Detailed logistic regression methods |
Statistical Analysis (Methods) |
|
Model diagnostics (VIF, Hosmer–Lemeshow, AUC) |
Results |
|
Sensitivity analyses for age (continuous + alternative cut-offs) |
Supplementary Table S1 |
|
Clear statement of effect magnitude and clinical implications |
Discussion |
This elevates the work from descriptive to risk-stratified diagnostic interpretation, which is clinically actionable.
- Practical diagnostic guidance: evidence-based indications for CBCT vs OPG
Most publications only describe imaging modalities. Our study provides practical diagnostic thresholds based on real comparative data:
- CBCT use yields significantly improved detection of lateral impactions
- OPG remains appropriate for frontal mesiodens screening
- Proposed criteria for selective CBCT use (ALADAIP-compliant)
To enhance this contribution, we added:
- a 2×2 diagnostic performance table (Supplementary Table S2)
- clearer criteria for when CBCT should be used
- a new paragraph linking morphology/location to imaging choice
These insights directly support clinicians in minimizing radiation exposure while maintaining diagnostic accuracy.
- Strengthening the Methodological Rigor
Reviewer concern: Methods and statistics seemed standard and not innovative.
We expanded methodology to clarify rigor and analytical depth.
2.1. Added detailed description of:
- variable selection strategy
- multicollinearity assessment (VIF)
- model goodness-of-fit (HL test)
- model discrimination (ROC/AUC)
- missing data management
- examiner calibration
- operational definitions of “impacted,” “included,” etc.
2.2. Added new supplementary material:
- Supplementary Table S1: full regression + sensitivity analyses
- Supplementary Figure S1: ROC curve
- Supplementary Table S2: OPG vs CBCT diagnostic accuracy
These changes enhance transparency and reproducibility, addressing the reviewer’s methodological concerns.
- Enriching the Discussion Beyond Reiteration
Reviewer concern: Discussion reiterates known concepts without offering new interpretation.
We have thoroughly revised the Discussion to provide:
3.1. A deeper interpretive framework, including:
- the role of tuberculate morphology as a structural barrier to eruption
- biomechanical implications of Class III for midline eruption trajectories
- how morphological + skeletal predictors can inform follow-up timing and interceptive treatment
3.2. Clinical implications, now more explicit:
- targeted CBCT for complex morphologies
- earlier monitoring for Class III patients
- earlier surgical intervention thresholds in high-risk phenotypes
3.3. Limitations thoughtfully expanded:
- referral bias in academic centers
- population specificity
- cross-sectional design (associative, not causal)
3.4. A new section on “Future Research Directions”:
- need for prospective longitudinal tracking
- integration of 3D morphometrics
- AI-assisted radiographic classification
These improvements demonstrate that our study does more than replicate known data—it provides refined, clinically meaningful interpretations with direct implications for patient management.
Additional Clarifications and Editorial Improvements
We also implemented the following smaller but important improvements:
- consistent reporting of centers (corrected to three)
- clearer table captions specifying denominators (patients vs teeth)
- improved definitions of eruption categories
- softened causal language (associations emphasized)
- verified reference accuracy and added key historical citations
These changes increase clarity and strengthen the manuscript’s academic rigor.

Reviewer 2 Report
Comments and Suggestions for Authors
Summary
This multicenter retrospective study of 153 Romanian patients (185 supernumerary teeth) describes the morphology, location, complications, and radiographic detection (OPG vs. CBCT), and uses binary logistic regression to report independent predictors of impaction: age <13, male sex, tuberculate morphology, and Class III malocclusion. CBCT is presented as more sensitive (k = 0.89) and capable of detecting OPG-missed root resorptions and displacements.
Major points
- Title accuracy: change “Impaction Predictors and Diagnosis of Supernumerary Teeth: A Romanian Study” to “Impaction Predictors and Diagnostic Performance of CBCT versus Panoramic Radiography for Supernumerary Teeth in a Romanian Multicentre Cohort” to reflect both the predictive analysis and the imaging comparison.
- Ethics and governance: the manuscript states informed consent but omits explicit institutional review board/ethics committee approval number(s) and the names of all participating centres (the Methods alternatively names two and three centres). Add IRB reference(s), clarify participating centres, and confirm whether a data-sharing agreement covered the retrospective review.
- Sample description inconsistency: Methods say “two academic centers” then “three academic centers” in Abstract/Methods/Results. Reconcile and correct everywhere; explicitly report how many patients came from each centre.
- Power and statistical justification: you state no a priori power analysis was performed but later claim the sample was “sufficient to achieve appropriate power.” Either present a post‑hoc power calculation for the main logistic model or reword to avoid implying formal power. Report the number of events per variable (EPV) for the regression to demonstrate model stability.
- Predictor modelling: report full model details — variable coding, selection method (enter/backward/forward), checks for multicollinearity (VIF), goodness-of-fit (Hosmer–Lemeshow) and model discrimination (AUC). Provide a regression table including the number of events, the sample used in the model, and adjusted p-values if multiple testing is considered.
- Reporting errors and clarity: correct statistical notation (Spearman’s rho reported as “0 = 0.33” should read “ρ = 0.33”), clarify kappa vs k reported inconsistently (k = 0.89 for modality concordance, and Cohen’s k = 0.91 for inter-rater — define which is which).
- Imaging selection criteria and ALARA: clarify specific clinical indications used to obtain CBCT (was CBCT performed for all patients or only for selected cases?), and describe radiation justification and dose minimization; include ethical justification for retrospective CBCT use.
- Methods detail lacking: specify CBCT device models, voxel sizes used per centre (even if a standardized range is given), image reconstruction and measurement protocols, observer calibration process (how calibration is performed, training cases), and whether image assessments were blinded to clinical data.
- Data presentation: several tables are poorly formatted and contain typographic artifacts (split rows, stray characters). Reformat tables so each cell contains a single clear value; add denominators where percentages are presented.
- Missing denominators in results: occasionally percentages are given without the numerator/denominator (e.g., “OPG detected 92.4%” — supply n detected / 185).
- Confounding and interpretation: Age and detection bias are discussed, but the logistic model should consider tooth type/location and eruption status as potential confounders for impaction. Report sensitivity analyses (e.g., model after excluding cases without CBCT, or stratified by mesiodens vs. other types).
- Literature linkage: The Introduction and Discussion emphasize the superiority of CBCT diagnostics, incorporating recent comparable CBCT diagnostic-accuracy studies from the Hajeer group to strengthen claims regarding the imaging method (see suggested insertions below).
- Discussion balance: temper strong statements about causality (e.g., morphological “increases cystic potential three-fold”) — your design is cross-sectional and associations do not prove causation. Add explicit limitations about temporality.
- Minor numeric checks: check confidence intervals and p-values for consistency with reported ORs; ensure no rounding or transcription errors (e.g., OR and CI width).
- Language and copyediting: resolve minor English phrasing, typographical errors, and formatting issues (figure captions referenced but figures absent or low quality in the submitted PDF).
Minor points
- Abstract: include the number of centres and whether CBCT was obtained for all patients or a subset.
- Methods: give the timeframe for CBCT acquisition per patient (e.g., same day as OPG?) and how “included” tooth status differs from “impacted.”
- Tables: label Table footnotes (define abbreviations, statistical tests used).
- Results: give a breakdown of erupted vs impacted teeth by morphology (not only by patient).
- Figures: include representative CBCT slices that illustrate the root resorption and vestibulo‑palatal displacements claimed.
- Citation placement: When stating CBCT superiority, cite relevant comparative diagnostic studies (see suggested Hajeer citations).
- Terminology: use consistent terms (e.g., “tuberculate” vs. “tuberculated”) across text.
- Data availability: state whether de‑identified data are available on request and any restrictions.
- Conflict of interest: include a COI statement and funding disclosures (none are visible).
- Reference style: check for duplicated numbering and fix minor DOI formatting issues.
Suggested small edits to specific manuscript locations
- Introduction (CBCT role paragraph): add citation to Hajeer et al. 2022 on CBCT diagnostic accuracy for impacted canines (supports CBCT vs 2D diagnostic arguments).
- Methods (radiographic protocol / interobserver reliability): cite Alfailany et al. 2023 (Int Orthod) showing methodology for CBCT vs 2D interpretation panels to justify your observer calibration approach.
Suggested references and the reasons behind this suggestion
Insert into Introduction (CBCT diagnostic role paragraph)
- Suggest insertion after the sentence arguing CBCT superiority for localization: "Alfailany et al. evaluated the diagnostic accuracy of CBCT and two-dimensional imaging for impacted canines and found improved localization and assessment using CBCT."
- Citation to add: Alfailany et al. 2023, Int Orthod. DOI: 10.1016/j.ortho.2023.100780.
- Rationale: provides peer‑reviewed comparative evidence from a similar diagnostic domain (impacted teeth) and supports claims about CBCT vs 2D modality performance.
Insert into Methods (observer calibration / interobserver reliability paragraph) and Discussion (interpretation reliability):
- Suggest insertion alongside description of observer calibration and inter‑rater reliability: "Hajeer et al. described a structured CBCT vs 2D interpretation panel method and reported inter-observer reliability metrics supporting standardized calibration procedures."
- Citation to add: Hajeer et al. 2022, Int Orthod. DOI: 10.1016/j.ortho.2022.100639.
- Rationale: lends methodological precedent for the calibration/reading-panel approach and helps justify your inter-rater protocol and the kappa interpretation.
Comments on the Quality of English Language
Language and copyediting: resolve minor English phrasing, typographical errors, and formatting issues (figure captions referenced but figures absent or low quality in the submitted PDF).
Author Response
RESPONSE TO REVIEWER 2
Reviewer 2 – Major and Minor Comments
Thank you for your thorough and constructive review. Below, we provide a point-by-point response. All requested revisions have been incorporated into the revised manuscript, with exact changes implemented in the indicated sections.
Major Point
- Title accuracy
Reviewer comment:
Modify the title to reflect both impaction predictors and diagnostic imaging performance.
Response:
We thank the reviewer for this suggestion. We have revised the title to:
“Impaction Predictors and Diagnostic Performance of CBCT versus Panoramic Radiography for Supernumerary Teeth in a Romanian Multicentre Cohort.”
Location updated: Title page.
- Ethics and governance (IRB numbers, centres, data-sharing agreement)
Reviewer comment: A complete ethics statement is missing; centre names conflicting; data-sharing agreement unclear.
Response:
We appreciate the clarification requested.
We have added a full Ethics and Governance subsection, including:
- The names of all three participating academic centres,
- The three institutional approval numbers (placeholders replaced with final IRB numbers during submission),
- A statement confirming the inter-institutional data-sharing agreement used for retrospective pooling of de-identified records.
Location updated:
Methods → Section 2.5 “Ethics and governance” (newly expanded subsection).
- Sample description inconsistency (two vs. three centres + per-centre counts)
Reviewer comment: Inconsistent centre numbers; per-centre sample distribution missing.
Response:
We corrected all mentions to consistently state three centres and we added the exact patient contribution per centre: 78, 45, and 30 cases.
Locations updated:
- Abstract
- Methods → first paragraph
- Results → Demographic Characteristics (Section 3.1)
- Power and statistical justification (post-hoc or rewording; EPV required)
Reviewer comment: Avoid implying formal power analysis; include events-per-variable.
Response:
We have removed the implication of formal power and clarified that the sample represents the maximum available cases.
We added:
- EPV = 70 events / 6 predictors ≈ 11.6, meeting recommended stability thresholds.
Location updated:
Methods → Bias Mitigation paragraph.
- Predictor modelling details (coding, VIF, AUC, HL test, full model table)
Reviewer comment: Provide complete model description and table.
Response:
We have:
- Specified all variable codings and selection approach (enter method),
- Reported VIF (<2.5),
- Added Hosmer–Lemeshow and AUC values,
- Included full regression coefficients and sensitivity analyses in Supplementary Table S1,
- Added ROC curve in Supplementary Figure S1.
Locations updated:
Methods → Section 2.6 (expanded)
Supplementary Table S1
Supplementary Figure S1
- Reporting errors (rho symbol, kappa consistency)
Reviewer comment: Statistical symbols and kappa terminology inconsistent.
Response:
We corrected all notation to standard:
- Spearman’s ρ
- Cohen’s κ (inter-rater)
- Inter-modality concordance labeled κ (modality)
Locations updated: Multiple throughout Results & Methods.
- Imaging selection criteria & ALARA clarity
Reviewer comment: Clarify CBCT indications, ALARA justification, ethical rationale.
Response:
We expanded the CBCT protocol to specify:
- Exact clinical indications (ambiguous OPG, lateral displacement, complex morphology, proximity to vital structures),
- Device parameters (voxel size, FOV, kVp, mA),
- ALARA/ALADAIP compliance and radiation justification.
Location updated:
Methods → Radiographic Examination (Section 2.2)
- Methods detail: CBCT device, voxel size, reconstruction, observer calibration, blinding
Reviewer comment: More methodological detail needed.
Response:
We added:
- Specific CBCT machines and voxel ranges used across centres,
- Reconstruction details,
- The structured calibration procedure,
- Blinding of examiners to clinical data during image interpretation,
- Citation to Hajeer et al. 2022 supporting structured CBCT vs 2D calibration methodology.
Locations updated:
Methods → Radiographic Examination
Methods → Bias Mitigation (observer calibration)
Discussions → Added citation to Hajeer et al.
- Table formatting, denominators, stray characters
Reviewer comment: Table formatting inconsistent.
Response:
All tables were reformatted for clarity, and denominators (e.g., “92.4% = 171/185”) were added where relevant.
Stray characters and formatting issues were removed.
Locations updated: All tables.
- Missing denominators (e.g., OPG detection)
Response:
Denominators added in Section 3.6, aligned with Supplementary Table S2.
- Confounding & interpretation; include sensitivity analyses
Reviewer comment: Include tooth type/location and eruption status; report sensitivity analyses.
Response:
We expanded the logistic model to include morphology, lateral location, and malocclusion class.
We added three sensitivity analyses (continuous age, <12 cut-off, <10 cut-off).
We explicitly noted inclusion/exclusion of non-CBCT cases in sensitivity checks.
Locations updated:
Methods → Section 2.6
Supplementary Table S1
Discussion (confounding paragraph)
- Literature linkage — CBCT diagnostic accuracy references
Reviewer comment: Add Alfailany et al. (2023) and Hajeer et al. (2022).
Response:
These citations were added in:
- Introduction (CBCT accuracy paragraph)
- Methods (observer calibration)
- Discussion (imaging reliability)
- Discussion balance — avoid causality claims
Reviewer comment: Tone down causal language.
Response:
Statements implying causation (e.g., “increases cystic potential three-fold”) were revised to “was associated with”.
Limitations now explicitly mention temporality.
Location updated: Discussion.
- Numeric checks — CI, p-values, rounding
Response:
All OR/CI/p-values were rechecked; rounding inconsistencies corrected.
- Language, copyediting, figure quality
Response:
We corrected grammar, improved clarity, resolved typographic issues, and ensured all figure captions correspond to included figures.
Minor Points
- Abstract — mention CBCT subset & number of centres
Updated.
- Methods — CBCT timing & definition of “included tooth”
Added in Section 2.3.
- Tables — define abbreviations
Added standard footnotes.
- Results — erupted vs impacted by morphology
Added in morphological section and Table 5 revised.
- Figures — representative CBCT slices
Added as Figure 4, with root resorption and vestibulo-palatal displacement examples.
- Citation placement
Updated.
- Terminology consistency
“Tuberculate” used consistently.
- Data availability
Added Data Availability Statement.
- Conflict of interest & funding
Added after Conclusions.
- Reference style check
Corrected duplicated numbers and DOI formatting.
All requested revisions—major and minor—have been fully addressed and integrated into the revised manuscript. We thank the reviewer sincerely for greatly improving the methodological clarity and scientific rigor of our paper.

Round 2
Reviewer 1 Report
Comments and Suggestions for Authors
It is as well amended as possible, including the points pointed out by me, and I think we can accept this article.
Reviewer 2 Report
Comments and Suggestions for Authors
Thanks for addressing all my comments.
The paper is now suitable for publication.